# An Adjoint-Free Algorithm for CNOPs via Sampling

Bin Shi[1,3] and Guodong Sun[2,3]

[1]State Key Laboratory of Scientific and Engineering Computing, Academy of Mathematics and Systems Science, Chinese Academy of Sciences, Beijing 100190, China
[2]State Key Laboratory of Numerical Modeling for Atmospheric Sciences and Geophysical Fluid Dynamics, Institute of Atmospheric Physics, Chinese Academy of Sciences, Beijing 100029, China
[3]University of Chinese Academy of Sciences, Beijing 100049, China

**Correspondence:** Bin Shi (Email: `shibin@lsec.cc.ac.cn`)

**Abstract.** In this paper, we propose a sampling algorithm based on state-of-the-art statistical machine learning techniques to obtain conditional nonlinear optimal perturbations (CNOPs), which is different from traditional (deterministic) optimization methods.[1] Specifically, the traditional approach is unavailable in practice, which requires numerically computing the gradient (first-order information) such that the computation cost is expensive since it needs a large number of times to run numerical models. However, the sampling approach directly reduces the gradient to the objective function value (zeroth-order information), which also avoids using the adjoint technique that is unusable for many atmosphere and ocean models and requires large amounts of storage. We show an intuitive analysis for the sampling algorithm from the law of large numbers and further present a Chernoff-type concentration inequality to rigorously characterize the degree to which the sample average probabilistically approximates the exact gradient. The experiments are implemented to obtain the CNOPs for two numerical models, the Burgers equation with small viscosity and the Lorenz-96 model. We demonstrate the CNOPs obtained with their spatial patterns, objective values, computation times and nonlinear error growth. Compared with the performance of the three approaches, all the characters for quantifying the CNOPs are nearly consistent, while the computation time using the sampling approach with fewer samples is extremely shorter. In other words, the new sampling algorithm shortens the computation time to the utmost at the cost of losing little accuracy.

## 1 Introduction

One of the critical issues for weather and climate predictability is the short-term behavior of a predictive model with imperfect initial data. For assessing subsequent errors in forecasts, it is of vital importance to understand the model's sensitivity to errors in the initial data. Perhaps the simplest and most practical way is to estimate the likely uncertainty in the forecast by considering to run with initial data polluted by the most dangerous errors. The traditional approach is the normal mode method (Rayleigh, 1879; Lin, 1955), which is based on the linear stability analysis and has been used to understand and analyze

---

[1]Generally, the statistical machine learning techniques refer to the marriage of traditional optimization methods and statistical methods, or says, stochastic optimization methods, where the iterative behavior is governed by the distribution instead of the point due to the attention of noise. Here, the sampling algorithm used in this paper is to numerically implement the stochastic gradient descent method, which takes the sample average to obtain the inaccurate gradient.

the observed cyclonic waves and long waves of middle and high latitudes (Eady, 1949). However, atmospheric and oceanic models are often linearly unstable. Specifically, the transient growth of perturbations can still occur in the absence of growing normal modes (Farrell and Ioannou, 1996a, b). Therefore, the normal mode theory is generally unavailable for the prediction generated by atmospheric and oceanic flows. To achieve the goal of prediction **on** a low-order two-layer quasi-geostrophic model in a periodic channel, Lorenz (1965) first proposed a non-normal mode approach and simultaneously introduced some new concepts, i.e., the tangent linear model, adjoint model, singular values, and singular vectors. Then, Farrell (1982) used this linear approach to investigate the linear instability within finite time. During the last decade of the past century, such a linear approach had been widely used to identify the most dangerous perturbations of atmospheric and oceanic flows, and also extended to explore error growth and predictability, such as patterns of the general atmospheric circulations (Buizza and Palmer, 1995) and the coupled ocean-atmosphere model of the El Niño-Southern Oscillation (ENSO) (Xue et al., 1997a, b; Thompson, 1998; Samelson and Tziperman, 2001). Recently, the non-normal approach had been extended further to an oceanic study for investigating the predictability of the Atlantic meridional overturning circulation (Zanna et al., 2011) and the Kuroshio path variations (Fujii et al., 2008).

Both the approaches of normal and non-normal modes are based on the assumption of linearization, which means that the initial error must be so small that a tangent linear model can approximately quantify the error's growth. Besides, the complex nonlinear atmospheric and oceanic processes have not yet been well considered in the literature. To overcome this limitation, Mu (2000) proposed a nonlinear non-normal mode approach with the introduction of two new concepts, nonlinear singular values and nonlinear singular vectors. Mu and Wang (2001) then used it to successfully capture the local fastest-growing perturbations for a 2D quasi-geostrophic model. However, several disadvantages still exist, such as unavailability in practice and unreasonable physics of the large norm for local fastest-growing perturbations. Starting from the perspective of nonlinear programming, Mu et al. (2003) proposed an innovative approach, named conditional nonlinear optimal perturbation (CNOP), to explore the optimal perturbation that can fully consider the nonlinear effect without any assumption of linear approximation. Generally, the CNOP approach captures initial perturbations with maximal nonlinear evolution given by a reasonable constraint in physics. Currently, the CNOP approach as a powerful tool has been widely used to investigate the fastest-growing initial error in the prediction of an atmospheric and oceanic event and to reveal some related mechanisms, such as the stability of the thermohaline circulation (Mu et al., 2004; Zu et al., 2016), the predictability of ENSO (Duan et al., 2009; Duan and Hu, 2016) and the Kuroshio path variations (Wang and Mu, 2015), the parameter sensitivity of the land surface processes (Sun and Mu, 2017), and typhoon-targeted observations (Mu et al., 2009; Qin and Mu, 2012). Some more details are shown in (Wang et al., 2020), and more perspectives on general fluid dyanmics in (Kerswell, 2018).

The primary goal of obtaining the CNOPs is to efficiently and effectively implement nonlinear programming, mainly including the spectral projected gradient (SPG) method (Birgin et al., 2000), sequential quadratic programming (SQP) (Barclay et al., 1998) and the limited memory Broyden-Fletcher-Goldfarb-Shanno (BFGS) algorithm (Liu and Nocedal, 1989) in practice. The gradient-based optimization algorithms are also adopted in the field of fluid dynamics to capture the minimal finite amplitude disturbance that triggers the transition to turbulence in shear flows (Pringle and Kerswell, 2010) and the maximal perturbations of the disturbance energy gain in a 2D isolated counter-rotating vortex-pair (Navrose et al., 2018). For the CNOP, the objective

function of the initial perturbations in a black-box model is obtained by the error growth of a nonlinear differential equation. Therefore, the essential difficulty here is how to efficiently compute the gradient (first-order information). Generally for an earth system model or an atmosphere-ocean general circulation model, it is unavailable to compute the gradient directly from the definition of numerical methods since it requires plenty of runs of the nonlinear model. Perhaps the most popular and practical way to numerically approximate the gradient is the adjoint technique, which is based on deriving the tangent linear model and the adjoint model (Kalnay, 2003). Once we can distill out the adjoint model, it becomes available to compute the gradient at the cost of massive storage space to save the basic state. In other words, the adjoint-based method takes a large amount of storage space to reduce computation time significantly. However, the adjoint model is unusable for many atmospheric and oceanic models since it is hard to develop, especially for the coupled ocean-atmosphere models. Still, the adjoint-based method can only deal with the smooth case. (Wang and Tan, 2010) proposed the ensemble-based methods, which introduces the classical techniques of EOF decomposition widely used in atmospheric science and oceanography. Specifically, it takes some principal modes of the EOF decomposition to approximate the tangent linear matrix. However, the colossal memory and repeated calculations occurring in the adjoint-based method still exist (Wang and Tan, 2010; Chen et al., 2015). In addition, the intelligent optimization methods (Zheng et al., 2017; Yuan et al., 2015) are unavailable on the high-dimension problem (Wang et al., 2020). All of the traditional (deterministic) optimization methods above cannot guarantee to find an optimal solution.

To overcome the limitations of the adjoint-based method described above, we start to take consideration from the perspective of stochastic optimization methods, which as the workhorse have powered recent developments in statistical machine learning (Bottou et al., 2018). In this paper, we use the stochastic derivative-free method proposed by Nemirovski and Yudin (1983, Section 9.3.2) that takes a basis on the simple high-dimensional divergence theorem (i.e., Stokes' theorem). With this stochastic derivative-free method, the gradient can be reduced to the objective function value in terms of expectation. For the numerical implementation as an algorithm, we take the sample average to approximate the expectation. Based on the law of large numbers, we provide a concentration estimate for the gradient by the general Chernoff-type inequality. This paper is organized as follows. The basic description of the CNOP settings and the proposed sampling algorithm are given in Section 2 and Section 3, respectively. We then perform the preliminary numerical test for two numerical models, the simple Burgers equation with small viscosity and the Lorenz-96 model in Section 4. A summary and discussion are included in Section 5.

## 2   The Basic CNOP Settings

In this section, we provide a brief description of the CNOP approach. It should be noted that the CNOP approach has been extended to investigate the influences of other errors, i.e., parameter errors and boundary condition errors, on atmospheric and oceanic models (Mu and Wang, 2017) beyond the original intention of CNOPs exploring the impact of initial errors. We only focus on the initial perturbations in this study.

Let $\Omega$ be a region in $\mathbb{R}^d$ with $\partial\Omega$ as its boundary. An atmospheric or oceanic model is governed by the following differential equation as

$$\begin{cases} \dfrac{\partial U}{\partial t} = F(U, P) \\ U|_{t=0} = U_0 \\ U|_{\partial\Omega} = G, \end{cases} \tag{1}$$

where $U$ is the reference state in the configuration space, $P$ is the set of model parameters, $F$ is a nonlinear operator, and $U_0$ and $G$ are the initial reference state and the boundary condition, respectively. Without loss of generality, we note $g^t(\cdot)$ to be the reference state evolving with time $U(t; \cdot)$ in the configuration space. Thus, given any initial state $U_0$, we can obtain that the reference state at time $T$ is $g^T(U_0) = U(T; U_0)$. If we consider the initial state $U_0 + u_0$ as the perturbation of $U_0$, then the reference state at time $T$ is given by $g^T(U_0 + u_0) = U(T; U_0 + u_0)$.

With both the reference states at time $T$, $g^T(U_0)$ and $g^T(U_0 + u_0)$, the objective function of the initial perturbation $u_0$ based on the initial condition $U_0$ is

$$J(u_0; U_0) = \left\| g^T(U_0 + u_0) - g^T(U_0) \right\|^2, ^2 \tag{2}$$

and then the CNOP formulated as the constrained optimization problem is

$$\max_{\|u_0\| \le \delta} J(u_0; U_0). \tag{3}$$

Both the objective function (2) and the optimization problem (3) come directly from the theoretical model (1). When we take the numerical computation, the properties of the two objects above, (2) and (3), probably become different. Here, it is necessary to mention some similarities and dissimilarities between the theoretical model (1) and its numerical implementation. If the model (1) is a system of ordinary differential equations, then it is finite-dimensional, and so there are no other differences between the theoretical model (1) and its numerical implementation except some numerical errors. However, if the model (1) is a partial differential equation, then it is infinite-dimensional. When we implement it numerically, the dimension is reduced to be finite for both the objective function (2) and the optimization problem (3). At last, we conclude this section with the notation $J(u_0)$ shortening $J(u_0; U_0)$ afterward for convenience.

## 3 Sample-based algorithm

In this section, we first describe the basic idea of the sampling algorithm. Then, it is shown for comparison with the baseline algorithms in numerical implementation. Finally, we conclude this section with a rigorous Chernoff-type concentration inequality to characterize the degree to which the sample average probabilistically approximates the exact gradient. The detailed proof is postponed to Appendix A.

The key idea for us to consider the sampling algorithm is based on the high-dimensional Stokes' theorem, which reduces the gradient in the unit ball to the objective value on the unit sphere in terms of the expectation. Let $\mathbb{B}^d$ be the unit ball in $\mathbb{R}^d$ and

$v_0 \sim \mathrm{Unif}(\mathbb{B}^d)$, a random variable $v_0$ following the uniform distribution in the unit ball $\mathbb{B}^d$. Given a small real $\epsilon > 0$, we can define the expectation of the objective function $J$ in the ball with the center $u_0$ and the radius $\epsilon$ as

$$\hat{J}(u_0) = \mathbb{E}_{v_0 \in \mathbb{B}^d} \left[ J(u_0 + \epsilon v_0) \right]. \tag{4}$$

In other words, the objective function $J$ is required to define in the ball $B(0; \delta + \epsilon) = \{u_0 \in \mathbb{R}^d : \|u_0\| \leq \epsilon + \delta\}$. Also, we find that $\hat{J}(u_0)$ is approximate to $J(u_0)$, that is, $\hat{J}(u_0) \approx J(u_0)$. If the gradient $\nabla J$ exists in the ball $B(0; \delta + \epsilon)$, the fact that the expectation of $v_0$ is zero tells us that the error of the objective value is estimated as

$$\|J(u_0) - \hat{J}(u_0)\| = O(\epsilon^2). \tag{}$$

Before proceeding to the next, we note the unit sphere as $\mathbb{S}^{d-1} = \partial \mathbb{B}^d$. With the representation of $\hat{J}(u_0)$ in (4), we can obtain the gradient $\nabla \hat{J}(u_0)$ directly from the function value $J$ by the high-dimensional Stokes' theorem as

$$\nabla \hat{J}(u_0) = \mathbb{E}_{v_0 \in \mathbb{B}^d} \left[ \nabla J(u_0 + \epsilon v_0) \right] = \frac{d}{\epsilon} \cdot \mathbb{E}_{v_0 \in \mathbb{S}^{d-1}} \left[ J(u_0 + \epsilon v_0) v_0 \right], \tag{5}$$

where $v_0 \sim \mathrm{Unif}(\mathbb{S}^{d-1})$ in the last equality. Similarly, $\nabla \hat{J}(u_0)$ is approximate to $\nabla J(u_0)$, that is, $\nabla \hat{J}(u_0) \approx \nabla J(u_0)$. If the gradient $\nabla J$ exists in the ball $B(0; \delta + \epsilon)$, we can show that the error of the gradient is estimated as

$$\|\nabla \hat{J}(u_0) - \nabla J(u_0)\| = O(d\epsilon). \tag{6}$$

The rigorous description and proof are shown in Appendix A (Lemma A.1 with its proof).

In the numerical computation, we obtain the approximate gradient, $\nabla \hat{J}(u_0)$, via the sampling as

$$\frac{d}{\epsilon} \cdot \mathbb{E}_{v_0 \in \mathbb{S}^{d-1}} \left[ J(u_0 + \epsilon v_0) v_0 \right] \approx \frac{d}{n\epsilon} \sum_{i=1}^{n} J(u_0 + \epsilon v_{0,i}) v_{0,i}, \tag{7}$$

where $v_{0,i} \sim \mathrm{Unif}(\mathbb{S}^{d-1})$, $(i = 1, \ldots, n)$ are the independent random variables following the identical uniform distribution on $\mathbb{S}^{d-1}$. Since the expectation of the random variable $v_0$ on the unit sphere $\mathbb{S}^{d-1}$, we generally take the following way with better performance in practice as

$$\frac{d}{\epsilon} \cdot \mathbb{E}_{v_0 \in \mathbb{S}^{d-1}} \left[ J(u_0 + \epsilon v_0) v_0 \right] = \frac{d}{\epsilon} \cdot \mathbb{E}_{v_0 \in \mathbb{S}^{d-1}} \left[ (J(u_0 + \epsilon v_0) - J(u_0)) v_0 \right] \approx \frac{d}{n\epsilon} \sum_{i=1}^{n} (J(u_0 + \epsilon v_{0,i}) - J(u_0)) v_{0,i}, \tag{8}$$

where $v_{0,i} \sim \mathrm{Unif}(\mathbb{S}^{d-1})$, $(i = 1, \ldots, n)$ are independent. From (8), $n$ is the number of samples and $d$ is the dimension. Generally in practice, the number of samples is far less than the dimension, $n \ll d$. Hence, the times to run the numerical model is $n + 1 \ll d + 1$, which is the times to run the numerical model via the definition of the numerical method as

$$\frac{\partial J(u_0)}{\partial u_{0,i}} \approx \frac{J(u_0 + \epsilon e_i) - J(u_0)}{\epsilon},$$

where $i = 1, \ldots, d$. For the adjoint method, the gradient is numerically computed as

$$\nabla J(u_0) \approx M^\top M u_0 \approx M^\top g^T (U_0 + u_0),$$

where $M$ is a product of some tangent linear models. Practically, the adjoint model, $M^T$, is hard to develop. In addition. we cannot obtain the tangent linear model for the coupled ocean-atmosphere models.

Next, we provide a simple but intuitive analysis of the convergence in probability for the samples in practice. With the representation of $\nabla \hat{J}(u_0)$ in (5), the weak law of large numbers states that the sample average (7) converges in probability toward the expected value, that is, for any $t > 0$, we have

$$\mathbf{Pr}\left(\left\|\frac{d}{n\epsilon}\sum_{i=1}^{n} J(u_0 + \epsilon v_{0,i})v_{0,i} - \nabla \hat{J}(u_0)\right\| \geq t\right) \to 0, \qquad \text{with} \quad n \to \infty.$$

Combined with the error estimate of gradient (6), if $t$ is assumed to be larger than $\Omega(d\epsilon)$ (i.e., there exists a constant $\tau > 0$ such that $t > \tau d\epsilon$), then the probability that the sample average approximates to $\nabla J(u_0)$ satisfies

$$\mathbf{Pr}\left(\left\|\frac{d}{n\epsilon}\sum_{i=1}^{n} J(u_0 + \epsilon v_{0,i})v_{0,i} - \nabla J(u_0)\right\| \geq t - \Omega(d\epsilon)\right) \to 0, \qquad \text{with} \quad n \to \infty.$$

Finally, we conclude the section with the rigorous Chernoff-type bound in probability for the simple but intuitive analysis above with the following theorem. The rigorous proof is shown in Appendix A with Lemma A.2 and Lemma A.3 proposed.

**Theorem 1.** If $J$ is continuously differentiable and satisfies the gradient Lipschitz condition, i.e., for any $u_{0,1}, u_{0,2} \in B(0,\delta)$, there exists a constant $L > 0$ such that the following inequality holds as

$$\|\nabla J(u_{0,1}) - \nabla J(u_{0,2})\| \leq L\|u_{0,1} - u_{0,2}\|.$$

For any $t > Ld\epsilon/2$, there exists a constant $C > 0$ such that the samples satisfy the concentration inequality as

$$\mathbf{Pr}\left(\left\|\frac{d}{n\epsilon}\sum_{i=1}^{n} J(u_0 + \epsilon v_{0,i})v_{0,i} - \nabla J(u_0)\right\| \geq t - \frac{Ld\epsilon}{2}\right) \leq 2\exp\left(-Cnt^2\right).$$

## 4 Numerical model and experiments

In this section, we perform several experiments to compare the proposed sampling algorithm with the baseline algorithms for two numerical models, the Burgers equation with small viscosity and the Lorenz-96 model. After the CNOP was first proposed in (Mu et al., 2003), plenty of methods, adjoint-based or adjoint-free, have been introduced to compute the CNOPs (Wang and Tan, 2010; Chen et al., 2015; Zheng et al., 2017; Yuan et al., 2015). However, some essential difficulties have still not been overcome. Taking the classical adjoint technique for example, the massive storage space and unusability in many atmospheric and oceanic modes are the two insurmountable points. In this study, different from traditional (deterministic) optimization methods above, we obtain the approximate gradient by sampling the objective values introduced in Section 3. Then, we use the second spectral projected gradient method (SPG2) proposed in (Birgin et al., 2000) to compute the CNOPs.

## 4.1 The Burgers equation with small viscosity

We first consider a simple theoretical model, the Burgers equation with small viscosity under the Dirichlet condition. It should be noted here that we adopt the internal units, $m$ and $s$. The reference state $U$ evolves nonlinearly with time as

$$\begin{cases} \dfrac{\partial U}{\partial t} + U\dfrac{\partial U}{\partial x} = \gamma\dfrac{\partial^2 U}{\partial x^2}, & (x,t) \in [0,L] \times [0,T] \\ U(0,t) = U(L,t) = 0, & t \in [0,T] \\ U(x,0) = \sin\left(\dfrac{2\pi x}{L}\right), & x \in [0,T] \end{cases} \tag{9}$$

where $\gamma = 0.005 m^2/s$ and $L = 100m$. We use the leapfrog/DuFort-Frankel scheme (i.e., the central finite difference scheme in both the temporal and spatial directions) to numerically solve the viscous Burgers equation above (9), with $\Delta x = 1m$ as the spatial grid size ($d = 101$) and $\Delta t = 1s$ as the time step. The objective function $J(u_0)$ used for optimization (2) can be rewritten in the form of the perturbation's norm square as

$$J(u_0) = \|u(T)\|^2 = \sum_{i=1}^{d} u_i(T)^2.$$

The constraint parameter is set to be $\delta = 8 \times 10^{-4} m/s$ such that the initial perturbation satisfies $\|u_0\| \leq \delta = 8 \times 10^{-4} m/s$. We note the numerical gradient computed directly as the definition method, where the step size for the difference is set to be $h = 10^{-8}$. Together with the adjoint methods, we set them as the baseline algorithms. For the sampling algorithm, we set the parameter $\epsilon = 10^{-8}$ in (4), the expectation of the objective function. The time $T$ in the objective function (2) is named as the prediction time. We take the two groups of numerical experiments according to the prediction time, $T = 30s$ and $T = 60s$, to calculate the CNOPs by the baseline algorithms and the sampling method. And then, we compare their performances to show the superiority of the sampling method.

The spatial distributions of the CNOPs computed by the baseline algorithms and the sampling method are shown in Figure 1, where we can find the change of the CNOPs' spatial pattern for the Burgers equation with small viscosity as:

(1) The spatial pattern of the CNOPs computed by two baseline algorithms, the definition method and the adjoint method, are nearly identical.

(2) Based on the spatial pattern of the CNOPs computed by two baseline algorithms, there are some fluctuating errors for the sampling method.

(3) When the number of samples increase from $n = 5$ to $n = 15$, the fluctuating errors in the spatial patter are reduced.

We have figured out the spatial distributions of the CNOPs in Figure 1. Or says, we have qualitatively characterized the CNOPs. However, we still need some quantities to measure the CNOPs' performance. Here, we use the objective value of the CNOP as the quantity. The objective values corresponding to the spatial patterns in Figure 1 are shown in Table 1. To show them clearly, we make them over that computed by the definition method with the percentage representation, which is

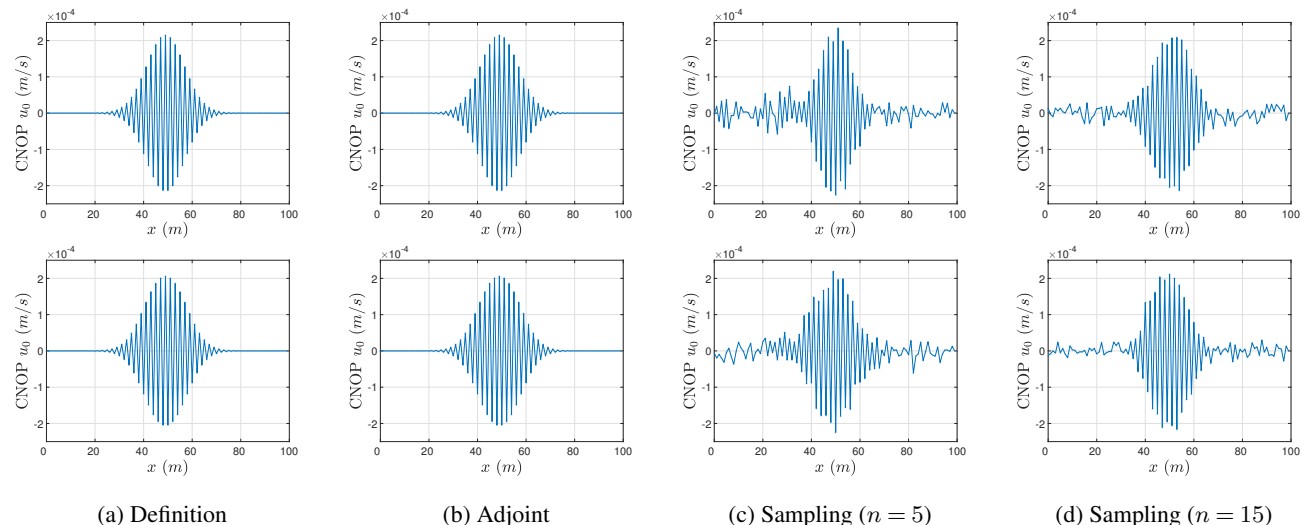

**Figure 1.** Spatial distributions of CNOPs (unit: $m/s$). Prediction time: on the top is $T = 30s$, and on the bottom is $T = 60s$.

| Method<br>Case | Definition | Adjoint | Sampling ($n = 5$) | Sampling ($n = 15$) |
|---|---|---|---|---|
| $T = 30s$ | $1.2351 \times 10^{-5}$ | $1.2351 \times 10^{-5}$ | $1.1727 \times 10^{-5}$ | $1.1950 \times 10^{-5}$ |
| | 100% | 100% | 94.95% | 96.75% |
| $T = 60s$ | 2.5035 | 2.5035 | 2.3899 | 2.4426 |
| | 100% | 100% | 95.46% | 97.57% |

**Table 1.** The objective values of CNOPs and the percentage over that computed by the definition method.

also shown in Table 1. For the Burgers equation with small viscosity, we can also find that the objective values by the adjoint method over that by the definition method is $100\%$ for both the two cases, $T = 30s$ and $T = 60s$, respectively; the objective values by the sampling method are all more than $90\%$; when we increase the number of samples from $n = 5$ to $n = 15$, the

percentage increases from $94.95\%$ to $96.75\%$ for the case $T = 30s$ and from $95.46\%$ to $97.57\%$ for the case $T = 60s$. The objective values of the CNOPs in Table 1 quantitatively echo the performances of spatial patterns in Figure 1.

    Next, we show the computation times to obtain the CNOPs by the baseline algorithms and the sampling method in Table 2. For the Burgers equation with small viscosity, the computation time taken by the adjoint method is far less than that directly by the definition method for the two cases, $T = 30s$ and $T = 60s$. When we implement the sampling method, the computation

time using $n = 15$ samples is almost the same as that taken directly by the definition method. If we reduce the number of samples from $n = 15$ to $n = 5$, the computation time is shortened by more than half.

    Finally, we describe the nonlinear evolution behavior of the CNOPs in terms of norm squares $\|u(t)\|^2$ computed by the baseline algorithms and the sampling method in Figure 2. We can find that there exists a fixed turning-time point, approximately

| Method / Case | Definition | Adjoint | Sampling ($n=5$) | Sampling ($n=15$) |
|---|---|---|---|---|
| $T=30s$ | $3.2788s$ | $1.0066s$ | **$0.3836s$** | $0.8889s$ |
| $T=60s$ | $6.3106s$ | $1.4932s$ | **$0.6464s$** | $1.4845s$ |

**Table 2.** Comparison of computation times (unit: $s$). Run on Matlab2022a with Intel® Core™ i9-10900 CPU@2.80GHz.

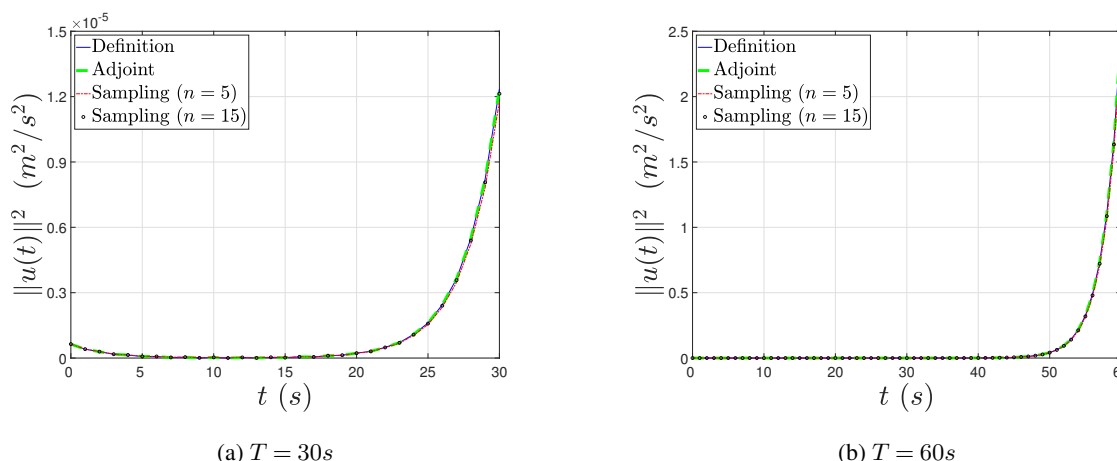

(a) $T = 30s$

(b) $T = 60s$

**Figure 2.** Nonlinear evolution behavior of the CNOPs in terms of the norm square.

$t = 20s$ for the prediction time $T = 30s$ and approximately $t = 50s$ for $T = 60s$. In the beginning, the nonlinear growth of the
CNOPs is very slow. When the evolving time comes across the fixed turning-time point, the perturbations start to proliferate.
Figure 2 shows the nonlinear evolution behaviors of the CNOPs computed by all the algorithms above are almost consistent
but do not provide any tiny difference between the baseline algorithms and the sampling method. So we further show that the
nonlinear evolution behavior of the CNOPs in terms of the difference $\Delta\|u(t)\|^2$ and relative difference $\Delta\|u(t)\|^2/\|u(t)\|^2$
based on the definition method in Figure 3. There is no difference or relative difference in the nonlinear error growth between
the two baseline algorithms. The top two graphs in Figure 3 show that the differences do not grow fast until the time comes
across the turning-time point. When we reduce the number of samples, the difference enlarges gradually, with the maximum
around $6 \times 10^{-7} m^2/s^2$ for $T = 30s$ and $0.12 m^2/s^2$ for $T = 60s$. However, the differences are very small compared with the
nonlinear growth of the CNOPs themselves, which is shown by the relative difference in the bottom two graphs of Figure 3.
In addition, some numerical errors exist around $t = 11s$ for the relative difference and decrease with increasing the number of
samples.

The Burgers equation with small viscosity is a partial differential equation, which is an infinite-dimensional dynamical sys-
tem. In the numerical implementation, it corresponds to the high-dimensional case. Taking all the performances with different
test quantities into account, i.e., spatial structures, objective values, computation times and nonlinear error growth, we conclude

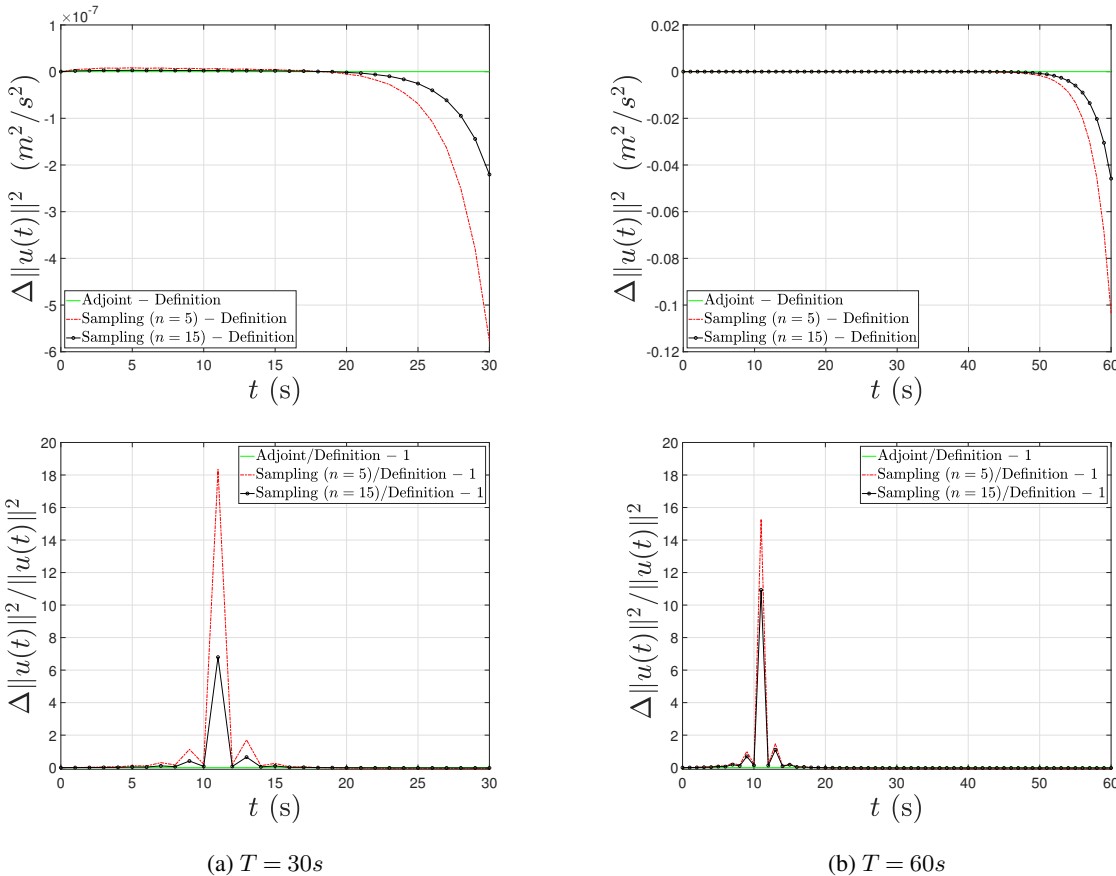

(a) $T = 30s$

(b) $T = 60s$

**Figure 3.** Nonlinear evolution behavior of the CNOPs in terms of the difference and relative difference of the norm square.

that the adjoint method obtains almost the total information and save much computation time simultaneously; the sampling method with $n = 15$ samples drops a few accuracies and loses little information but share nearly the same computation time; when we reduce the number of samples from $n = 15$ to $n = 5$, we can obtain about $95\%$ of information as the baseline algorithms, but the computation time is reduced more than half. The cause for the phenomenon described above is perhaps due to the high-dimensional property.

### 4.2 The Lorenz-96 model

Next, we consider the Lorenz-96 model, one of the most classical and idealized models, which is designed to study fundamental issues regarding the predictability of the atmosphere and weather forecasting (Lorenz, 1996; Lorenz and Emanuel, 1998). In recent two decades, the Lorenz-96 model has been widely applied in data assimilation and predictability (Ott et al., 2004; Trevisan and Palatella, 2011; De Leeuw et al., 2018) to studies in spatiotemporal chaos (Pazó et al., 2008).The Lorenz-96

model is also used to investigate the predictability of extreme amplitudes of travelling waves (Sterk and van Kekem, 2017),
which points out that it depends on the dynamical regime of the model.

With a cyclic permutation of the variables as $\{x_i\}_{i=1}^{N}$ satisfying $x_{-1} = x_{N-1}$, $x_0 = x_N$, $x_1 = x_{N+1}$, the governing system
of equations for the Lorenz-96 model is described as

$$\frac{dx_i}{dt} = \underbrace{(x_{i+1} - x_{i-2})\, x_{i-1}}_{\text{advection}} - \underbrace{x_i}_{damping} + \underbrace{F}_{\text{external forcing}} \tag{10}$$

where the system is nondimensional. The physical mechanism considers that the total energy is conserved by the advection,
decreased by the damping and kept away from zero by the external forcing. The variables $x_i$ $(i = 1, \ldots, N)$ can be interpreted
as values of some atmospheric quantity (e.g., temperature, pressure or vorticity) measured along a circle of constant latitude
of the earth (Lorenz, 1996). The Lorenz-96 model (10) can also describe waves in the atmosphere. Specifically, Lorenz (1996)
observed that the waves slowly propagate westward toward decreasing $i$ for $F > 0$ sufficiently large.

In this study, we use the classical 4th-order Runge-Kutta method to numerically solve the Lorenz-96 model (10). The

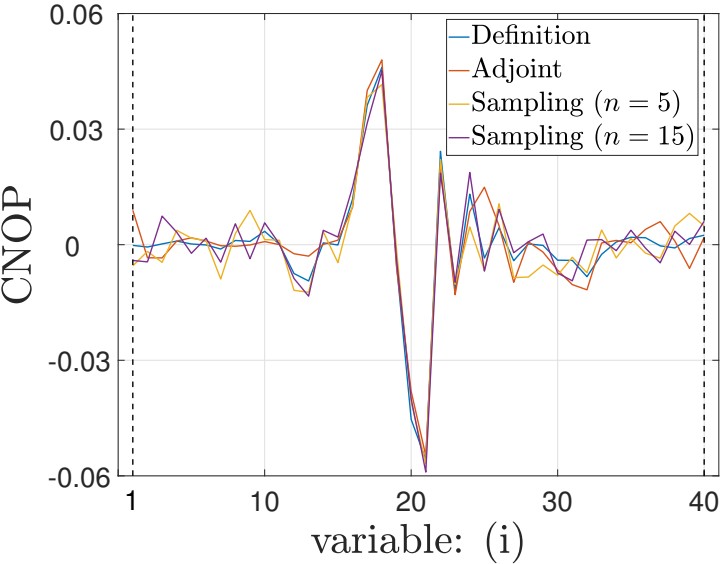

**Figure 4.** Spatial distributions of CNOPs.

two parameters are set as $N = 40$ and $F = 8$, respectively. The spatial distributions of the CNOPs computed by the baseline
algorithms and the sampling method are shown in Figure 4. Unlike the three dominant characters described above for the
Burgers equation with small viscosity, the spatial distributions of the CNOPs computed by all four algorithms are almost
consistent except for some little fluctuations for the Lorenz-96 model. Similarly, we provide a display for the objective values
of the CNOPs computed by the baseline algorithms and the sampling method and the percentage over that computed by the
definition method in Table 3. We find that the percentage of the objective value computed by the adjoint method is only $92.35\%$,

less than that by the sampling method for the Lorenz-96 model. In addition, the difference between the number of samples $n = 5$ and $n = 15$ is only $0.57\%$ in the percentage of the objective value. In other words, we can obtain about $95\%$ of the total information by taking the sampling algorithm only using $n = 5$ samples.

| Definition | Adjoint | Sampling ($n = 5$) | Sampling ($n = 15$) |
|---|---|---|---|
| 50.9099 | 47.0154 | **48.0157** | 48.3093 |
| 100% | 92.35% | **94.32%** | 94.89% |

**Table 3.** The objective values of CNOPs and the percentage over that computed by the definition method.

Similarly, we show the computation times to obtain the CNOPs by the baseline algorithms and the sampling method in Table 4. For the adjoint method, different from the Burgers equation with small viscosity, the time to compute the CNOP by the adjoint method is almost twice that used by the definition method for the Lorenz-96 model. However, the computation time that the sampling method uses is less than $1/3$ of what the definition method uses. When we reduce the number of samples from $n = 15$ to $n = 5$, the computation time decreases by more than $0.1s$. As a result, the sampling method only using $n = 5$ samples saves much computation time to obtain the CNOP for the Lorenz-96 model.

| Definition | Adjoint | Sampling ($n = 5$) | Sampling ($n = 15$) |
|---|---|---|---|
| $3.5576s$ | $6.6346s$ | **$0.9672s$** | $1.0829s$ |

**Table 4.** Comparison of computation times. Run on Matlab2022a with Intel® Core™ i9-10900 CPU@2.80GHz.

Finally, we demonstrate the nonlinear evolution behavior of the CNOPs in terms of norm squares $\|x(t)\|^2$ computed by the baseline algorithms and the sampling method in Figure 5. Recall Figure 2 for the Burgers equation with small viscosity, the norm squares of the CNOPs have almost no growth until the turning-time point and then proliferate. Unlikely, the norm squares of the CNOPs almost grow linearly for the Lorenz-96 model without any turning-time point. Similarly, since the four nonlinear evolution curves of norm squares almost coincide in Figure 5, we cannot find any tiny difference in the nonlinear growth of the CNOPs between the baseline algorithms and the sampling method. So we still need to observe the nonlinear evolution behavior of the CNOPs in terms of the difference $\Delta\|x(t)\|^2$, which is shown in the left panel of Figure 6. In the initial stage, the three nonlinear evolution curves share the same growth behavior, with the maximum amplitude being the one by implementing the sampling method with $n = 5$ samples. Afterward, the error's amplitude decreases for the sampling method with $n = 5$ samples, and the two curves by the adjoint method and the sampling method with $n = 5$ samples are similar, with the larger amplitude being the one by the adjoint method, which achieves the maximum around $0.3$. Indeed, the differences are very small compared with the nonlinear growth of the CNOPs themselves, which is shown by the relative difference $\Delta\|x(t)\|^2/\|x(t)\|^2$ in the right panel of Figure 6. We can find that the three curves of the relative difference share the same nonlinear evolution behavior, with the sampling method of different numbers of samples on both sides and the adjoint method in between. When the number

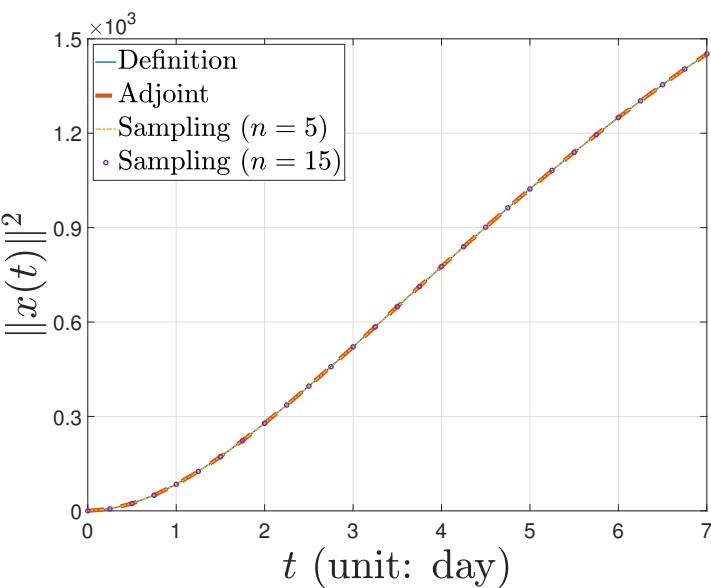

**Figure 5.** Nonlinear evolution behavior of the CNOPs in terms of the norm square.

of samples is reduced, the amplitude of the relative difference decreases. In addition, the order of the relative difference's magnitude is $10^{-3}$, which is so tiny that there are no essential differences.

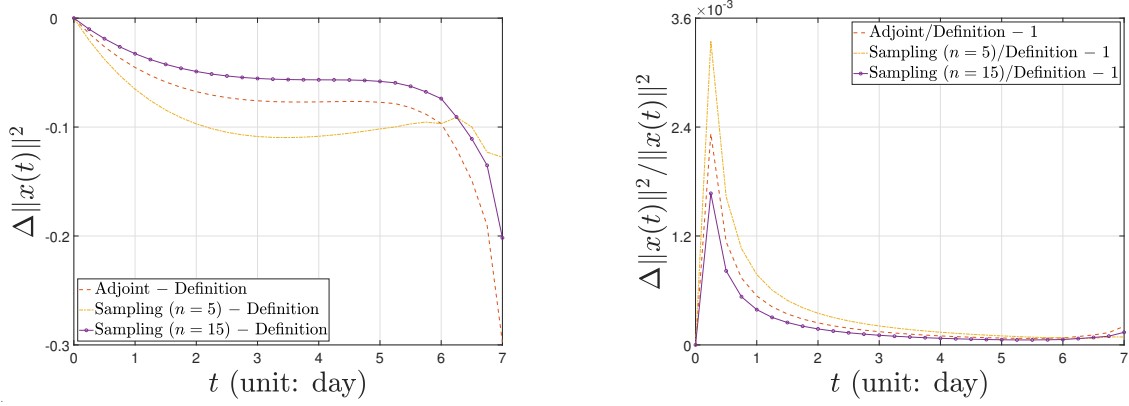

**Figure 6.** Nonlinear evolution behavior of the CNOPs in terms of the difference and relative difference of the norm square.

Although the dimension of the Lorenz-96 model is not very large due to being composed of a finite number of ordinary differential equations, it possesses strongly nonlinear characters. Unlike the Burgers equation with small viscosity, the adjoint method does not work well for the Lorenz-96 model, which spends more computation time and obtains less percentage of the total information. The sampling method performs more advantages in the computation, saving far more computation time

and obtaining more information. However, the performance in reducing the number of samples from $n = 15$ to $n = 5$ is not obvious. Perhaps this is due to the characters of the Lorenz-96 model, strong nonlinearity and low dimension.

## 5 Summary and discussion

In this paper, we introduce a sampling algorithm to compute the CNOPs based on the state-of-the-art statistical machine learn-ing techniques. The theoretical guidance comes from the high-dimensional Stokes' theorem and the law of large numbers. We
derive a Chernoff-type concentration inequality to rigorously characterize the degree to which the sample average probabilis-tically approximates the exact gradient. We show the advantages of the sampling method by comparison with the performance of the baseline algorithms, e.g., the definition method and the adjoint method. If there exists the adjoint model, the computation time is reduced significantly with the exchange of much storage space. However, the adjoint model is unusable for the complex atmospheric and oceanic model in practice.
For the numerical tests, we choose two simple but representative models, the Burgers equation with small viscosity and the Lorenz-96 model. The Burgers equation with small viscosity is one of the simplest nonlinear partial differential equations sim-plified from the Navier-Stokes equation, which holds a high-dimensional property. The Lorenz-96 model is a low-dimensional dynamical system with strong nonlinearity. For the numerical performance of a partial differential equation, the Burgers equa-tion with small viscosity, we find that the adjoint method performs very well and saves much computation time; the sampling
method can share nearly the same computation time with the adjoint method with dropping a few accuracies by adjusting the number of samples; the computation time can be shortened more by reducing the number of samples further with the nearly consistent performance. For the numerical performance of a low-dimensional and strong nonlinear dynamical system, the Lorenz-96 model, we find that the adjoint method takes underperformance, but the sampling method fully occupies the dominant position, regardless of saving the computation time and performing the CNOPs in terms of the spatial pattern, the
objective value and the nonlinear growth. Still, unlike the Burgers equation with small viscosity, the performance is not obvious for reducing the number of samples for the Lorenz-96 model. Based on the comparison above, we propose a possible conclu-sion that the sampling method probably works very well for an atmospheric or oceanic model in practice, which is a partial differential equation with strong nonlinearity. Perhaps the high efficiency of the sampling method performs more dominant, and the computation time is shortened obviously by reducing the number of samples.
Currently, the CNOP method has been widely applied to predictability in meteorology and oceanography. For the nonlinear multiscale interaction (NMI) model (Luo et al., 2014, 2019), an atmospheric blocking model which successfully characterizes the life cycle of the dynamic blocking from onset to decay, the CNOP method has been used to investigate the sensitivity on initial perturbations and the impact of the westerly background wind (Shi et al., 2022). However, it is still very challenging to compute the CNOP for a realistic earth system model, such as the Community Earth System Model (CESM) (Wang et al.,
2020). Many difficulties still exist, even for a high-regional resolution model, such as the Weather Research and Forecasting (WRF) Model, which is used widely in operational forecasting (Yu et al., 2017). Based on increasingly reliable models, we now comment on some extensions for further research to compute and investigate the CNOPs on the more complex models by

the sampling method, regardless of theoretical or practical. An idealized ocean-atmosphere coupling model, the Zebiak-Cane (ZC) model (Zebiak and Cane, 1987), which might characterize the oscillatory behavior of ENSO in amplitude and period based on oceanic wave dynamics. Mu et al. (2007) computed the CNOP of the ZC model by use of its adjoint model to study the spring predictability barrier for El Niño events. In addition, Mu et al. (2009) also computed the CNOP of the PSU/NCAR mesoscale model (i.e., the MM5 model) using its adjoint model to explore the predictability of tropical cyclones. It looks very interesting and practical to test the validity of the sampling algorithm to calculate the CNOPs on the two more realistic numerical models, the ZC model and the MM5 model, as well as the idealized NMI model. For an earth system model or atmosphere-ocean general circulation models (AOGCMs), it is often unavailable to obtain the adjoint model, so the sampling method provides a probable way of computing the CNOPs to investigate its predictability. In addition, it becomes possible for us to use 4D-Var data assimilation on a coupled climate system model when the sampling method is introduced. Also, it is thrilling to implement the sampling method in the Flexible Global Ocean-Atmosphere-Land System (FGOALS)-s2 (Wu et al., 2018) to make the decadal climate prediction.

*Code availability.* The codes that support the findings of this study are available from the corresponding author, [author initials], upon reasonable request.

## Appendix A: Proof of Theorem 1

**Lemma A.1.** If the expectation $\hat{J}(u_0)$ is given by (4), then the expression (5) is satisfied. Also, under the same assumption of Theorem 1, the difference between the expectation of objective value and itself can be estimated as

$$\|\hat{J}(u_0) - J(u_0)\| \leq \frac{L\epsilon^2}{2};$$  (A1)

and the difference between the expectation of gradient and itself can be estimated as

$$\|\nabla\hat{J}(u_0) - \nabla J(u_0)\| \leq \frac{Ld\epsilon}{2}.$$  (A2)

*Proof of Lemma A.1.* First, with the definition of $\hat{J}(u_0)$, we show the proof of (5), the equivalent representation of the gradient $\nabla\hat{J}(u_0)$.

- For $d = 1$, the gradient of the expectation $\hat{J}$ about $u_0$ can be computed as

$$\frac{d\hat{J}(u_0)}{du_0} = \frac{d}{du_0}\left(\frac{1}{2}\int_{-1}^{1} J(u_0 + \epsilon v_0)dv_0\right) = \frac{1}{2}\int_{-1}^{1}\frac{dJ(u_0 + \epsilon v_0)}{\epsilon dv_0}dv_0 = \frac{J(u_0 + \epsilon) - J(u_0 - \epsilon)}{2\epsilon}.$$

- For the case of $d \geq 2$, we assume that $\mathbf{a} \in \mathbb{R}^d$ is an arbitrary vector. Then, the gradient $\nabla \hat{J}(u_0)$ satisfies the following equality as

$$
\begin{aligned}
\boldsymbol{a} \cdot \nabla \hat{J}(u_0) &= \int_{v_0 \in \mathbb{B}^d} \boldsymbol{a} \cdot \nabla_{u_0} J(u_0 + \epsilon v_0) dV \\
&= \frac{1}{\epsilon} \int_{v_0 \in \mathbb{B}^d} \nabla_{v_0} \cdot (J(u_0 + \epsilon v_0) \boldsymbol{a}) dV \\
&= \frac{1}{\epsilon} \int_{v_0 \in \mathbb{S}^{d-1}} J(u_{0,k} + \epsilon v_0) \boldsymbol{a} \cdot v_0 dS \\
&= \boldsymbol{a} \cdot \frac{1}{\epsilon} \int_{v_0 \in \mathbb{S}^{d-1}} J(u_0 + \epsilon v_0) v_0 dS.
\end{aligned}
$$

Because the vector $\boldsymbol{a}$ is arbitrary, we can obtain the following equality:

$$
\nabla \int_{v_0 \in \mathbb{B}^d} J(u_0 + \epsilon v_0) dV = \frac{1}{\epsilon} \int_{v_0 \in \mathbb{S}^{d-1}} J(u_0 + \epsilon v_0) v_0 dS.
$$

Since the ratio of the surface area and the volume of the unit ball $\mathbb{B}^d$ is $d$, the equivalent representation of the gradient (5) is satisfied.

If the objective function $J$ is continuously differentiable and satisfies the gradient Lipschitz condition, we can obtain the following inequality as

$$
\left| J(u_0 + \epsilon v_0) - J(u_0) - \epsilon \langle \nabla J(u_0), v_0 \rangle \right| \leq \frac{L\epsilon^2}{2} \|v_0\|^2.
$$

Because $\int_{v_0 \in \mathbb{B}^d} \langle \nabla J(u_0), v_0 \rangle \, dV = 0$, the estimate (A1) is obtained directly.

- For any $i \neq j \in \{1, \ldots, d\}$, since $v_{0,i}$ and $v_{0,j}$ are uncorrelated, we have

$$
\int_{v_0 \in \mathbb{S}^{d-1}} v_{0,i} v_{0,j} dS = 0.
$$

- For any $i = j \in \{1, \ldots, d\}$, we have

$$
\int_{v_0 \in \mathbb{S}^{d-1}} v_{0,i}^2 dS = \frac{1}{d} \int_{v_0 \in \mathbb{S}^{d-1}} \left( \sum_{i=1}^d v_{0,i}^2 \right) dS = \frac{1}{d} \int_{v_0 \in \mathbb{S}^{d-1}} dS.
$$

Since $v_0$ is a row vector, we derive the following equality as

$$
\mathbb{E}_{v_0 \in \mathbb{S}^{d-1}}[v_0^T v_0] = \frac{1}{d} \cdot \mathbf{I}.
$$

Hence, we obtain the equivalent representation of the gradient $\nabla J(u_0)$ as

$$
\nabla J(u_0) = \frac{d}{\epsilon} \cdot \mathbb{E}_{v_0 \in \mathbb{S}^{d-1}} \left[ \epsilon \langle \nabla J(u_0), v_0 \rangle v_0 \right].
$$

Finally, since $v_0 \sim \text{Unif}(\mathbb{S}^{d-1})$, then $\mathbb{E}_{v_0 \in \mathbb{S}^{d-1}}[v_0] = 0$. Hence, we estimate the difference between the expectation of gradient and itself as

$$
\begin{aligned}
\|\nabla \hat{J}(u_0) - \nabla J(u_0)\| &\leq \left\| \frac{d}{\epsilon} \cdot \mathbb{E}_{v_0 \in \mathbb{S}^{d-1}}\left[(J(u_0 + \epsilon v_0) - J(u_0))\, v_0\right] - \frac{d}{\epsilon} \cdot \mathbb{E}_{v_0 \in \mathbb{S}^{d-1}}\left[\epsilon \langle \nabla J(u_0), v_0 \rangle v_0\right] \right\| \\
&\leq \frac{d}{\epsilon} \cdot \mathbb{E}_{v_0 \in \mathbb{S}^{d-1}}\left[\left\| J(u_0 + \epsilon v_0) - J(u_0) - \epsilon \langle \nabla J(u_0), v_0 \rangle \right\| \cdot \|v_0\|\right] \\
&\leq \frac{L d \epsilon}{2},
\end{aligned}
$$

where the last inequality follows the gradient Lipschitz condition.

$\square$

Considering any $\epsilon > 0$, to proceed with the concentration inequality, we still need to know that the random variable $J(u_0 + \epsilon v_0)$ for $v_0 \sim \text{Unif}(\mathbb{S}^{d-1})$ is sub-Gaussian. Thus, we first introduce the following lemma.

**Lemma A.2** (Proposition 2.5.2 in Vershynin (2018)). Let $X$ be a random variable. If there exist two constants $K_1, K_2 > 0$ such that the moment generating function of $X^2$ is bounded:

$$
\mathbb{E}\left[\exp\left(\frac{X^2}{K_1^2}\right)\right] \leq K_2,
$$

then the random variable $X$ is sub-Gaussian.

Because $J(u_0 + \epsilon v_0)$ is bounded on $\mathbb{S}^{d-1}$, $\exp(J(u_0 + \epsilon v_0)^2 / K_1^2)$ is integrable on $\mathbb{S}^{d-1}$ for any $K_1 > 0$, i.e., there exists a constant $K_2 > 0$ such that

$$
\mathbb{E}_{v_0 \in \mathbb{S}^{d-1}}\left[\exp\left(\frac{J(u_0 + \epsilon v_0)^2}{K_1^2}\right)\right] \leq K_2.
$$

With Lemma A.2, the random variable $J(u_0 + \epsilon v_0)$ is sub-Gaussian. Therefore, for any fixed vector $v_0' \in \mathbb{S}^{d-1}$, we know the random variable $J(u_0 + \epsilon v_0)\langle v_0, v_0' \rangle$ is sub-Gaussian. We now introduce the following lemma to proceed with the concentration inequality.

**Lemma A.3** (Theorem 2.6.3 in Vershynin (2018)). Let $X_1, \ldots, X_n$ be independent, mean zero, sub-Gaussian random variables, and $a = (a_1, \ldots, a_n) \in \mathbb{R}^n$. Then, for every $t \geq 0$, we have

$$
\mathbf{Pr}\left(\left|\sum_{i=1}^n a_i X_i\right| \geq t\right) \leq 2\exp\left(-\frac{ct^2}{K^2 \|a\|^2}\right),
$$

where $K = \max_{1 \leq i \leq n} \|X_i\|_{\psi_2}$.[3]

---

[3]The sub-Gaussian norm of a random variable $X$ is defined as

$$
\|X\|_{\psi_2} = \inf\left\{t > 0 : \mathbb{E}\exp\left(\frac{X^2}{t^2}\right) \leq 2\right\}.
$$

Combined with Lemma A.1 and Lemma A.3, we can obtain the concentration inequality for the samples as

$$\mathbf{Pr}\left(\left|\frac{d}{n\epsilon}\sum_{i=1}^{n}\langle J(u_0+\epsilon v_{0,i})v_{0,i}v_0'\rangle - \langle\nabla\hat{J}(u_0),v_0'\rangle\right| \geq t\right) \leq 2\exp\left(-\frac{cnt^2}{K^2}\right),$$

where $v_0'$ is any unit vector on $\mathbb{S}^{d-1}$. Thus we can proceed with the concentration estimate by the Cauchy-Schwarz inequality as

$$\mathbf{Pr}\left(\left\|\frac{d}{n\epsilon}\sum_{i=1}^{n}J(u_0+\epsilon v_{0,i})v_{0,i} - \nabla\hat{J}(u_0)\right\| \geq t\right) \leq 2\exp\left(-\frac{cnt^2}{K^2}\right).$$

Based on the triangle inequality, we can proceed with the concentration inequality with the estimate of the difference between the expectation of objective value and itself (A2) as

$$\mathbf{Pr}\left(\left\|\frac{d}{n\epsilon}\sum_{i=1}^{n}J(u_0+\epsilon v_{0,i})v_{0,i} - \nabla J(u_0)\right\| \geq t - \frac{Ld\epsilon}{2}\right) \leq 2\exp\left(-\frac{cnt^2}{K^2}\right)$$

for any $t > Ld\epsilon/2$. Taking $C = c/K^2$, we complete the proof of Theorem 1.

*Author contributions.* Bin Shi constructed the basic idea of this paper, derived all formulas and the proofs, coded the sampling method in Matalb to show all the figures, and wrote the manuscript. Guodong Sun joined the discussions of this manuscript and provided some suggestions. All the authors contributed to the writing and reviewing of the manuscript.

*Competing interests.* The authors declare that they have no conflict of interest.

*Acknowledgements.* We are indebted to Mu Mu for seriously reading an earlier version of this paper and providing his suggestions about this theoretical study. Bin Shi would also like to thank Ya-xiang Yuan, Ping Zhang, and Yu-hong Dai for their encouragement to understand and analyze the nonlinear phenomena in nature from the perspective of optimization in the early stages of this project. This work was supported by Grant No.12241105 of NSFC and Grant No.YSBR-034 of CAS.

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
