# Peer review of "An Adjoint-Free Algorithm for CNOPs via Sampling"

_EGUsphere, 2022_

## Referee Comment (RC1)

**Review of "An Adjoint-Free Algorithm for CNOPs via Sampling"**

The three main problems in calculating CNOP are: long computation times, large data storage and adjoint model development. This study tries to use the sample-based method to compute CNOP. A notable feature of the sample-based approach is that it is an adjoint-free algorithm and requires less storage. Theoretically, this method may be useful for obtaining CNOP. However, this study is too preliminary to present enough information about the practical application of sample-based method. So the current version of the manuscript may not be accepted unless more valid information is given.

**Major Comments:**

1.    The sample-based algorithm is very similar to the ensemble-based method (Wang and Tan, MWR, 2010). It is necessary to compare these two approaches. I also find that the ensemble-based result (Fig. 2 in Wang and Tan, 2010) seems to have less error than the sample-based algorithm.

2.    Line 105, this equation is satisfied if the gradient exists. As mentioned in section 1, the gradient may not exist in some cases, such as the on-off process. In this situation, whether the sample-based method can be used to calculate the CNOP. If not, it will be a great challenge to use this method in practical models.

3.    What are the similarities between the CNOPs obtained by the sample-based methods with gradient-definition or adjoint approaches? In the Burgers model, to what extent the value of $n$ affects the accuracy of the CNOPs? Could it be possible to provide the results with larger $n$ values?

4.    Are the results sensitive to the choice of the samples? How to choose the samples, especially for the realistic applications?

5.    The Burgers equations is too simple. To show the valid of the method, I strongly recommend using a GCM realistic model to test this method. In fact, in the GCM model, the sample number may be very important. Larger sample number may be required in realistic model. But too large sample number will limit the potential use of this method, because too many nonlinear model runs need to be performed. How to obtain the accurate CNOP with suitable sample number?

6.    Fig. 3. Why are the errors remarkable larger when t=11?

**Minor Comments:**

1. Line 60: "faults" is inappropriate.
2. Line 62: "an optimal solution" can also hardly be guaranteed in other approaches, such as ensemble or even adjoint approaches in GCM model.
3. Line 145 $\epsilon = 10^{-8}$ is selected for calculating the CNOPs. Does the value of this

parameter affect the obtained CNOPs?

4. Line 150 Further clarify the parameter $\alpha = 10^{-8}$. Is it the same with $\epsilon = 10^{-8}$

5. Line 189: "We implement…" →"We will implement…"

Wang, B. and Tan, X. 2010: Conditional nonlinear optimal perturbations: Adjoint-free calculation method and preliminary test, Monthly Weather Review, 138, 1043–1049

---

## Author Response (AR3)

**The First Response to Reviewers' Comments**

We thank the editor and the two referees for thoroughly reading the manuscript and their helpful comments. We are very pleased to see many positive remarks. For example, the first reviewer said, "A notable feature of the sample-based approach is that it is an adjoint-free algorithm and requires less storage." Reviewer 2 added, "The authors propose to determine the gradient by the Monte-Carlo method, which may be more economical than the adjoint method (but apparently at the cost of lower accuracy). That is perfectly all right. The mathematical method is clearly defined and mathematically sound." In light of the comments, we have made a thorough revision addressing all major concerns, resulting in a significantly improved paper version.

**1 The First Reviewer's Report**

**1.1 Major Comments**

1. *The sample-based algorithm is very similar to the ensemble-based method [Wang and Tan, 2010]. It is necessary to compare these two approaches. I also find that the ensemble-based result (Fig. 2 in [Wang and Tan, 2010]) seems to have less error than the sample-based algorithm.*

   Thank you for pointing out that it is necessary to compare the sample-based algorithm with the ensemble-based method [Wang and Tan, 2010].

   **The ensemble-based method**  The classical adjoint method to compute the CNOP is composed of two parts, computing the tangent linear matrix and obtaining the adjoint matrix, or the representation of adjoint iteration, by transposing the tangent linear matrix. The ensemble-based method [Wang and Tan, 2010] introduces the techniques of the EOF decomposition widely used in atmospheric science and oceanography, which takes some principal modes to approximate the tangent linear matrix. In other words, the ensemble-based method [Wang and Tan, 2010] converts the computation of the adjoint matrix into the computation EOF decomposition, where the large amount of storage and repeated calculations occurring in the adjoint method still exist [Wang et al., 2020]. Furthermore, the ensemble-based method [Wang and Tan, 2010] is an algorithm under the framework of the adjoint-based method. Meanwhile, the choice of localization radius is also empirical.

   **The sample-based method**  The sample-based algorithm has no relationship with the tangent linear matrix and the adjoint matrix. The gradient is first approximated by the expectation in the small ball-type neighborhood $O(u_0, \epsilon)$. By Stokes' theorem, the expectation in the ball can be transformed into that on the sphere. Then, we take the samples with the average to approximate the expectation. The samples following the uniform distribution are chosen to be fully independent, which has no dependence on the numerical model and the objective function.

   Hence, we conclude that the sample-based algorithm is essentially different from the ensemble-based method [Wang and Tan, 2010].

2. *Line 105, this equation is satisfied if the gradient exists. As mentioned in Section 1, the gradient may not exist in some cases, such as the on-off process. In this situation, whether the sample-based method can be used to calculate the CNOP. If not, it will be a great challenge to use this method in practical models.*

Yes, the sample-based method can be used to calculate the CNOP. Actually, we compute the gradient approximately following the way as

$$\nabla \hat{J}(u_0) = \frac{d}{\epsilon} \cdot \mathbb{E}_{v_0 \in \mathbb{S}^{d-1}}[J(u_0 + \epsilon v_0)v_0] \approx \frac{d}{n\epsilon} \sum_{i=1}^{n} J(u_0 + \epsilon v_{0,i})v_{0,i},$$

where only the function value is used in the computation. The unique difference is that if the gradient exists, it is easy for us to get the estimate in Line 105; if the gradient does not exist, we need to characterize the first-order information to obtain the estimate in Line 105, i.e., the subgradient for the continuous objective function which is not differentiable.

3. *What are the similarities between the CNOPs obtained by the sample-based methods with gradient-definition or adjoint approaches? In the Burgers model, to what extent the value of $n$ affects the accuracy of the CNOPs? Could it be possible to provide the results with larger $n$ values?*

Thanks for these good comments. The similarities between the CNOPs obtained by the sample-based methods with gradient-definition or adjoint approaches are that they are very similar in **the spatial distributions (patterns)** in Figure 1 (the paper). Specifically, the CNOP obtained by the sample-based method is very similar to that by the baseline algorithms, the definition method and the adjoint method, except for some fluctuations generated by the noise in **the spatial distribution**. However, we still need a quantity to characterize the similarities. In the revised version, we add the objective function values generated by the CNOPs obtained as the testing quantity, which is shown in Table 1.

| Case \ Method | Definition | Adjoint | Sampling ($n = 5$) | Sampling ($n = 15$) |
|---|---|---|---|---|
| $T = 30s$ | $1.2351 \times 10^{-5}$ | $1.2351 \times 10^{-5}$ | $1.1727 \times 10^{-5}$ | $1.1950 \times 10^{-5}$ |
| | 100% | 100% | 94.95% | 96.75% |
| $T = 60s$ | 2.5035 | 2.5035 | 2.3899 | 2.4426 |
| | 100% | 100% | 95.46% | 97.57% |

Table 1: The objective values of CNOPs and the percentage over that computed by the definition method.

In Table 1, we can find that the percentage increase when we increase the number of samples. In other words, with the increase of the number of samples, the CNOP computed by the sampling method is closer to that by the baseline algorithm. Here, we add case $n = 30$ to show the number of samples affect the accuracy of the CNOPs in Figure 1. The spatial distributions of the CNOPs corresponding to Figure 1 is shown in Table 2. We can also find

| Case \ Sampling | $n = 5$ | $n = 15$ | $n = 30$ |
|---|---|---|---|
| $T = 30s$ | $1.1727 \times 10^{-5}$ 94.95% | $1.1950 \times 10^{-5}$ 96.75% | $1.2212 \times 10^{-5}$ 98.87% |
| $T = 60s$ | $2.3899$ 95.46% | $2.4426$ 97.57% | $2.4738$ 98.81% |

Table 2: The objective values of CNOPs and the percentage over that computed by the definition method.

the fluctuations decrease and the spatial pattern is closer to that by the defention method with increasing samples in Figure 1

[Figure]

(a) $n = 5$      (b) $n = 15$      (c) $n = 30$

Figure 1: Spatial distributions of CNOPs $(m/s)$. Prediction time: on the top is $T = 30s$, and on the bottom is $T = 60s$.

4. *Are the results sensitive to the choice of the samples? How to choose the samples, especially for the realistic applications?*

   In the two experiments for two numerical models, the Burgers equation with small viscosity and the Lorenz-96 model, the results are not sensitive when the number of samples are no less than $n = 5$. In this study, we generate the samples in Matlab by $v \sim \text{Unif}(\mathbb{S}^{d-1})$, the random variable following the uniform distribution on the unit sphere. We believe that it should follow the same way and be not hard for the realistic applications in the GCM models.

5. *The Burgers equations is too simple. To show the valid of the method, I strongly recommend using a GCM realistic model to test this method. In fact, in the GCM model, the sample*

*number may be very important. Larger sample number may be required in realistic model. But too large sample number will limit the potential use of this method, because too many nonlinear model runs need to be performed. How to obtain the accurate CNOP with suitable sample number?*

Thanks for the suggestion. In this paper, we propose the sampling method to obtain the CNOPs, focusing on the theoretical investigation. Under your suggestion, we add the Lorenz-96 model, one of the most classical models used to investigate the predictability of the atmosphere and weather forecasting, to further investigate the sampling method on the strong nonlinear model. The performance shows the superiority of the sampling method, saving more computation time and obtaining about 95% of total information. In the next step, we will start to investigate the sampling method to obtain the CNOPs on more realistic models, such as the Zebiak-Cane and MM5 models. Finally, we will implement the sampling method to obtain the CNOPs on a realistic earth system model or atmosphere-ocean general circulation model. Meanwhile, we will introduce the sampling method into the 4D-Var data assimilation.

6. *Fig. 3. Why are the errors remarkable larger when $t = 11$?*

Thanks for the careful observation. The errors are remarkably larger around $t = 11$ on the nonlinear evolution of relative difference, because the relative error computed by the definition method evolving at $t = 11$ is so close to zero that the small deviation generated by the sampling method over it becomes very large. It is verified that the relative error's decrease is very obvious, when we reduce the number of samples from $n = 15$ to $n = 5$.

**1.2 Minor Comments**

1. *Line 60: "faults" is inappropriate.*

   Thanks, we have taken "disadvantages" instead of "faults".

2. *Line 62: "an optimal solution" can also hardly be guaranteed in other approaches, such as ensemble or even adjoint approaches in GCM model.*

   Thanks for pointing the point out. All of the traditional (deterministic) optimization methods above can not guarantee to find an optimal solution. We have revised the sentenses below line 62, which is in the revised version from Line 15 to Line 70.

3. *Line 145: $\epsilon = 10^{-8}$ is selected for calculating the CNOPs. Does the value of this parameter affect the obtained CNOPs?*

   We select the parameter $\epsilon = 10^{-8}$ for calculating the CNOPs. Actually, when we select the order of magnitude no more than $O(10^{-6})$, the CNOPs obtained are almost not affected by the parameter $\epsilon$. The obtained CNOPs has some observed errors when we take the parameter $\epsilon = 10^{-5}$.

4. *Line 150: Further clarify the parameter $\alpha = 10^{-8}$. Is it the same with $\epsilon = 10^{-8}$?*

Thanks for the suggestion. To make it clear, we modify the notation $\alpha$ as $h$ to strengthen that the parameter is used as the step size to compute the number gradient as

$$\nabla J(u) \approx \frac{J(u + h e_i) - J(u)}{h}$$

where $e_i$ $(i = 1, \ldots, N)$ is the unit orthogonal basis with the $i$th entry being one and others zero. The parameter of step size $h = 10^{-8}$ used to compute the number gradient is different from $\epsilon = 10^{-8}$, which is used to sampling the gradient as

$$\nabla \hat{J}(u_0) = \mathbb{E}_{v_0 \in \mathbb{B}^d} \left[ \nabla J(u_0 + \epsilon v_0) \right] = \frac{d}{\epsilon} \cdot \mathbb{E}_{v_0 \in \mathbb{S}^{d-1}} \left[ J(u_0 + \epsilon v_0) v_0 \right] \approx \frac{d}{n\epsilon} \sum_{i=1}^{n} J(u_0 + \epsilon v_{0,i}) v_{0,i}.$$

5. *Line 189: "We implement ..."* $\longrightarrow$ *"We will implement ..."*

Thanks for pointing out the grammar mistake. We have rewritten the last section. The new sentense locates on Line 300. It looks very interesting and practical to test the validity of the sampling algorithm to calculate the CNOPs on the two more realistic numerical models, the ZC model and the MM5 model.

**2 The Second Reviewer's Report**

**2.1 Major Comments**

1. *The initial state defined by equation (7) is antisymmetric with respect to the middle point $x = L/2$ of the spatial domain $[0, L]$, and will remain so in the ensuring evolution. The advective velocity $U$ is positive for $x < L/2$, and negative for $x > L/2$. This means that the advection term $U \partial U / \partial x$ will cause convergence of the flow onto the middle point. The viscosity term will counteract that convergence, and one can expect that the flow will evolve to an asymptotic equilibrium state. A simple calculation actually shows that an asymptotic state to be*
$$U_\infty(x) = 2\gamma \left( \frac{\pi}{L} \right) \tan \left( \frac{\pi x}{L} \right).$$
*It has a singularity at point $x = L/2$, where the velocity $U$ becomes infinite and changes sign. The dynamics that result from equation (7) is simply an evolution of the flow towards the asymptotic equilibrium under the opposite effects of advection and viscosity. That is a very singular situation, from which no conclusion of some generality can be drawn.*

Thanks for the good comment. We appreciate the reviewer's sharp intuition in physics. Indeed, the singular phenomenon that the velocity $U$ becomes infinite will appear when the time that we implement the numerical simulation for the Burgers equation with small visocity is long enough. However, what we consider here is the finite prediction time. In other words, the nonlinear evolution behavior of the Burgers equation at $T = 30s$ or $T = 60s$ is far from the blow-up. Hence, it is reasonable for us to compute the CNOP at $T = 30s$ or $T = 60s$ for the Burgers equation with small visocity with the special initial condition.

2. *Clearly, the authors are aware that their paper is not significant from a geophysical point of view, and they present in their conclusion a number of further experiments they intend to perform with realistic atmospheric and/or oceanic numerical models. These experiments sound perfectly reasonable, and I can only encourage the authors to implement their projects. But even sooner than that, I suggest they perform experiments with the model defined by Lorenz [1996] (see also [Lorenz and Emanuel, 1998]). That model, in addition of being of small dimension (40 in its most common use) and easy to integrate, possesses a number of basic properties that make it a good reference for geophysical applications: (1) it contains both a physical forcing and an internal dissipation, the competition of which controls the energetics of the system; (2) it contains nonlinear energy-conserving interactions, somewhat similar to nonlinear advection, which make the flow chaotic; (3) it has nonlinear waves somewhat similar to atmospheric waves.*

Thanks for the great suggestion. We are encouraged to add some numerical experiments for the Lorenz-96 model from a geophysical viewpoint in the revised manuscript. Please refer to Section 4.2 in the revised manuscript.

**2.2 Minor Comments**

1. *The authors stress that their method does not require the use of an adjoint. They write (abstract, l. 4) that **the adjoint technique** [. . .] **produces instability due to linearization,** with similar statements at other places in the paper. They do not give any supporting evidence for that. For what I know, the adjoint technique is on the contrary very robust and reliable. For instance, the European Centre for Medium-Range Weather Forecasts (ECMWF) has used the adjoint approach since 1997, with a complete meteorological model, for its daily operational system of variational assimilation, without ever encountering problems of stability (a fairly large number of other meteorological or oceanographical centres have also used the adjoint approach for long periods without difficulties). It may be that the authors refer to situations where the gradient with respect to the control variable (e.g., as here, the initial condition) varies so rapidly with that variable that the gradient is useless for numerical applications (such situations will occur for instance when one considers gradients with respect to initial conditions of a chaotic system, if the prediction time is large enough for the celebrated butterfly effect to have significantly impacted the prediction). But that is totally independent of how the gradient is determined, whether by the adjoint method or by another method. Unless the authors can give specific examples, I suggest they remove any reference to instabilities resulting from the linearization introduced in the adjoint approach.*

We appreciate the great suggestion. We have removed all the content about **instabilities resulting from the linearization** introduced in the adjoint approach. Instead, we add that **the adjoint-based model is unusable for many atmospheric and oceanic models since the adjoint models are not easy to develop**. In addition, we also find that the adjoint method does not perform very well in the Lorenz-96 model, spending more computation but the performance in spatial pattern and objective values less than that computed by the sample method.

2. *Concerning again the adjoint method, the authors write (ll. 58-59) **a large amount of***

*storage space to save the basic state during each iteration is a critical issue, that produces [...]* **long computation times.** *It is precisely because it avoids large computing times that the adjoint approach has been implemented in many applications. But it is true that the corresponding economy in time is obtained at the cost of a large amount of storage space (in any operational system, saving computing time matters fundamentally much more than saving storage space)*

Thanks for pointing this out. We know that the adjoint method is an important advance in computing the CNOPs. In the new revised version, we have modified the sentenses from Line 59 to line 64.

> Perhaps the most popular and practical method adopted to obtain the gradient is the adjoint technique, which is based on calculating the tangent linear model and the adjoint matrix [Kalnay, 2003]. Specifically, when we can distill out the adjoint matrix or the expression of adjoint iteration, it becomes available to compute the gradient at the cost of massive storage space to save the basic state. In other words, the adjoint-based method uses a large amount of storage space to exchange the significant reduction of computation time.

3. *L. 146, ...* **we perform numerical experiments to calculate the CNOPs directly from the definition,** *... (with similar statements in other places, see also Figure 1). How can one determine the CNOPs directly from the definition (equation 3) ?*

Thanks for pointing this out. The definition method indicates that the gradient is computed by the definition of the numerical method as

$$\nabla J(u_0)_i \approx \frac{J(u_0 + \epsilon e_i) - J(u_0)}{\epsilon}$$

where $i = 1, \ldots, d$ and $e_i$ is a vector with the $i$th entry being one and the other being zero.

4. *The English is occasionally faulty or difficult to understand. In particular, the last line of the abstract (l. 10) makes no sense to me.*

Thanks for the suggestion on the English expression. We have removed the last sentence in the abstract. We have also reorganized the language and rewritten the abstract, introduction, and the last section – summary and discussion. And we have revised the other three sections.

**References**

E. Kalnay. *Atmospheric modeling, data assimilation and predictability.* Cambridge university press, 2003.

E. N. Lorenz. Predictability: A problem partly solved. In: Proc. Seminar on Predictability. *ECMWF Reading, Berkshire, UK*, 1:1–18, 1996.

E. N. Lorenz and K. A. Emanuel. Optimal sites for supplementary weather observations: Simulation with a small model. *Journal of the Atmospheric Sciences*, 55(3):399–414, 1998.

B. Wang and X. Tan. Conditional nonlinear optimal perturbations: Adjoint-free calculation method and preliminary test. *Monthly Weather Review*, 138(4):1043–1049, 2010.

Q. Wang, M. Mu, and G. Sun. A useful approach to sensitivity and predictability studies in geophysical fluid dynamics: conditional non-linear optimal perturbation. *National Science Review*, 7(1):214–223, 2020.

**The Second Response to Reviewers' Comments**

We thank the editor and the two referees for thoroughly reading the manuscript and for their helpful comments. We are very pleased to see many positive remarks. After we have addressed his/her concerns, the first referee recommended accepting this manuscript as it is. Also, we are happy to see that the new referee said "I see a lot of potential in this approach". In light of the comments, we have made a thorough revision addressing all major concerns, resulting in a significantly improved paper version.

**1 Main concerns of the third referee**

1. *It is said in the abstract that machine learning approaches are used in the current manuscript. I do not see any place where such tools are used, unless I am mistaken. This is misleading for the reader. This should be revisited in the whole manuscript.*

   Thanks for pointing out this problem. We have added the explanation in the revised manuscript. Followed by the first sentence in the abstract (with the superscript label 1 (**line 3**)), we add the explanation for machine learning approaches as

   - Generally, the statistical machine learning techniques refer to the marriage of traditional optimization methods and statistical methods, or says, stochastic optimization methods, where the iterative behavior is governed by the distribution instead of the point due to the attention of noise. Here, the sampling algorithm used in this paper is to numerically implement the stochastic gradient descent method, which takes the sample average to obtain the inaccurate gradient

2. *In Section 3, the method is outlined based on some demonstration of theorems, but there is no information on how the approach is implemented and compared with traditional methods. Please revisit completely this aspect in order to show the way to follow in order to develop the method by others.*

   Thanks for the good suggestion. We have added the content on how the approach is implemented and compared with traditional methods in **Section 3**. (**Line 130**)

   - In the numerical computation, we obtain the approximate gradient, $\nabla \hat{J}(u_0)$, via the sampling as
     $$\frac{d}{\epsilon} \cdot \mathbb{E}_{v_0 \in \mathbb{S}^{d-1}} \left[ J(u_0 + \epsilon v_0) v_0 \right] \approx \frac{d}{n\epsilon} \sum_{i=1}^{n} J(u_0 + \epsilon v_{0,i}) v_{0,i}, \tag{1.1}$$
     where $v_{0,i} \sim \mathrm{Unif}(\mathbb{S}^{d-1})$, $(i = 1, \ldots, n)$ are the independent random variables following the identical uniform distribution on $\mathbb{S}^{d-1}$. Since the expectation of the random variable $v_0$ on the unit sphere $\mathbb{S}^{d-1}$, we generally take the following way with better performance

in practice as

$$\frac{d}{\epsilon} \cdot \mathbb{E}_{v_0 \in \mathbb{S}^{d-1}} \left[ J(u_0 + \epsilon v_0) v_0 \right] = \frac{d}{\epsilon} \cdot \mathbb{E}_{v_0 \in \mathbb{S}^{d-1}} \left[ \left( J(u_0 + \epsilon v_0) - J(u_0) \right) v_0 \right]$$

$$\approx \frac{d}{n\epsilon} \sum_{i=1}^{n} \left( J(u_0 + \epsilon v_{0,i}) - J(u_0) \right) v_{0,i}, \qquad (1.2)$$

where $v_{0,i} \sim \text{Unif}(\mathbb{S}^{d-1})$, $(i = 1, \dots, n)$ are independent. From (1.2), $n$ is the number of samples and $d$ is the dimension. Generally in practice, the number of samples is far less than the dimension, $n \ll d$. Hence, the times to run the numerical model is $n + 1 \ll d + 1$, which is the times to run the numerical model via the definition of the numerical method as

$$\frac{\partial J(u_0)}{\partial u_{0,i}} \approx \frac{J(u_0 + \epsilon e_i) - J(u_0)}{\epsilon},$$

where $i = 1, \dots, d$. For the adjoint method, the gradient is numerically computed as

$$\nabla J(u_0) \approx M^\top M u_0 \approx M^\top g^T (U_0 + u_0),$$

where $M$ is a product of some tangent linear models. Practically, the adjoint model, $M^T$, is hard to develop. In addition. we cannot obtain the tangent linear model for the coupled ocean-atmosphere models.

3. *I have really hard time to read and understand the English. It should considerably be improved to make the results understandable.*

Thanks. We have improved considerable places for the English expression in the revised version, with bold text as a marker for editor and reviewer to track the changes.